# Accuracy of COVID-19 relevant knowledge among youth: Number of information sources matters

**Patricia Wonch Hill**[1]*, **Judy Diamond**[2], **Amy N. Spiegel**[3], **Elizabeth VanWormer**[4], **Meghan Leadabrand**[1], **Julia McQuillan**[1]

**1** Department of Sociology, University of Nebraska, Lincoln, Nebraska, United States of America,
**2** University Libraries & State Museum, University of Nebraska, Lincoln, Nebraska, United States of America,
**3** Department of Educational Psychology, University of Nebraska, Lincoln, Nebraska, United States of America, **4** School of Veterinary Medicine & Biomedical Sciences, School of Natural Resources, University of Nebraska, Lincoln, Nebraska, United States of America

* phill3@unl.edu

## Abstract

Can comics effectively convey scientific knowledge about COVID-19 to youth? What types and how many sources of information did youth have about COVID-19 during the pandemic? How are sources of information associated with accurate COVID-19 knowledge? To answer these questions, we surveyed youth in grades 5–9 in a Midwestern United States school district in the winter of 2020–2021. The online survey used measures of COVID-19 knowledge and sources, with an embedded experiment on COVID-19 relevant comics. Guided by an integrated *science capital* and *just-in-time health and science information acquisition* model, we also measured level of science capital, science identity, and utility of science for health and society. The school district protocol required parental consent for participation; 264 of ~15,000 youth participated. Youth were randomly assigned one of four comic conditions before receiving an online survey. Results indicate that, similar to knowledge gains in comic studies on other science topics, reading the comics was associated with 7 to 29% higher accuracy about COVID-19. We found that youth reported getting information about COVID-19 from between 0–6 sources including media, family, friends, school, and experts. The bivariate positive association of news versus other sources with accuracy of knowledge did not persist in the full model, yet the positive association of a higher number of sources and accuracy did persist in the multivariate models. The degree of valuing the utility of science for their health moderated the number of sources to accuracy association. Those with less value on science for health had a stronger positive association of number of sources and accuracy in COVID-19 knowledge. We conclude that during a pandemic, even with health and science information ubiquitous in the news media, increasing youth access to a variety of accurate sources of information about science and health can increase youth knowledge.

**Data Availability Statement:** All data files are available from the Zenodo database (DOI: https://zenodo.org/deposit/6461729).

**Funding:** This study was supported by the 2020 National Science Foundation grant, RAPID DRL2028026: Using Popular Media to Educate Youth About the Biology of Viruses and the Current COVID-19 Pandemic (JD, JM, LVW, PWH) (nsf. gov). Parts of this publication were also supported by the National Institute of General Medical Sciences at the National Institutes of Health under award R25GM129836 (JM, PWH) (nihsepa.org). Any opinions, findings, or conclusions expressed in this material are those of the authors and do not necessarily reflect the views of the National Science Foundation or the National Institutes of Health. The funders had no role in study design, data collection and analysis, decision to publish, or preparation of the manuscript.

**Competing interests:** The authors have declared that no competing interests exist.

## Introduction

Beginning in the spring of 2020, young people in the United States (U.S.) were confronted with how to understand the Coronavirus Disease 2019 (COVID-19) and the virus SARS-CoV-2. In the U.S., directed health measures aiming to "flatten the curve" disrupted the lives of every person, especially youth [1–4]. During the first year of the pandemic, many youth and their families dealt with closed schools and had to navigate online teaching and learning [5]. The novelty of the disease meant that scientists, public health officials, and the public (i.e., youth and adults) were all learning about COVID-19 at the same time. Unlike most science and health information, COVID-19 was often breaking news and became politicized in the U. S. [6]. Research on adult and youth media consumption has shown that a large percentage of people may actively avoid media about news and politics in the context of seemingly infinite media consumption choices [7,8]. Unlike adults, youth may experience passive exposure to information at school or through media they share with their parents/guardians (e.g. radio, TV) [7]. Youth may also actively seek out information that interests them from their own sources, or they may live in media rich environments with high science capital where families communicate and engage one another about health and science topics [7,9,10].

The COVID-19 pandemic was unprecedented in several ways, so it is unclear if prior research on sources of science information would be relevant. Family and household members likely learn about COVID-19 from similar sources, yet youth may also learn from people and media that are unique or even tailored to them [7]. Youth oriented information could come from peers, school, teachers, social media, other websites, and marketing aimed at engaging youth audiences. Unlike most adults, many students are also exposed to relatively up-to-date information about science and health through their school courses [10]. Families may participate in some science-related activities together (e.g., visiting science museums, zoos, and nature centers); but youth often have more opportunities to engage with science in youth-focused clubs, summer camps, and afterschool enrichment programs compared to adults [11–13].

Two theories are particularly useful for guiding research on youth acquisition of accurate science and health information: *science capital* [11,14] and *just-in-time science and health information acquisition system* (JIT) [15]. First, building on Bourdieu's theory of social capital, Archer et al. [11,14] developed a theory of science capital to describe the differential access youth have to science experiences through their parents, schools, and communities. The unequal access to science capital contributes to inequalities among youth in science identity in early adolescence and shapes attitudes about the utility of science for their lives, their health, their future careers, and for society [14,16]. Second, Miller et al.'s JIT theory focuses on the motivations for seeking out health and science information [15]. JIT focuses on adult information acquisition and argues that curiosity drives science information consumption, while health information is motivated by salience and utilitarian reasons. Integrating these two theories is important to understand youth access to and acquisition of science and health information, particularly during the COVID-19 pandemic.

The social capital and JIT theories were developed based on adults, therefore it is useful to consider the application to youth. Generally, adults and youth in the U.S. have relatively low health and science literacy, and lower ability to assess media sources (i.e., media literacy) compared to other Western countries [12,13]. Even if youth use similar sources of information as adults they are unlikely have the same interpretations as adults [17,18]. Human development studies indicate that children process information differently than adolescents and adults [19]. Scientific-ways-of-knowing develop and change as youth mature; younger children often struggle with abstractions and the idea that other people may have different perspectives than their own [20,21]. Youth centered media campaigns and health and science materials tailored

to youth can help to make accurate health and science information more accessible to youth, and even more equitable [22–24].

There is evidence that conveying accurate information in comics appeals to youth with lower science identities, enabling them to learn accurate science information from sources that are more engaging than the kinds of essays often provided in textbooks [25]. By integrating art with a narrative structure and fictional storyline, comics can make abstract and complex concepts more understandable [26–28]. These fictional texts have the potential to make science learning more equitable through the inclusion of characters and situations that readers may connect with their own lives, accessible visualizations, and narratives that are naturally more approachable and memorable than explanative text [22,26,29].

We know that the type and number of sources of information matter for COVID-19 knowledge for adults [30] but whether they matter for youth is unclear. In addition, little is understood about how youth science capital and perceptions of the utility of science for health might explain the association between sources of information, science and health information acquisition, and accuracy of knowledge. Finally, little information exists about how providing youth with accurate information about COVID-19 (i.e., using youth-targeted media like comics) might impact their knowledge. To address these gaps in the literature, the overall aim of this work was to learn where middle-school-aged youth got information about COVID-19 during the pandemic, and to investigate if the type or number of sources were associated with accuracy of COVID-19 relevant knowledge. We also wanted to investigate if any association was explained by youth science capital, science identity, or attitudes about the utility of science for health or society. We found that, after controlling for demographic characteristics, learning environments, science capital, science identity, and attitudes about the utility of science for health and society, the number of sources of information is positively associated with accuracy of COVID-19 knowledge, particularly for youth who have lower attitudes about the utility of science for health. This finding indicates that there is a need for accurate science information that is appealing to youth of all backgrounds in order to fill gaps in youth understanding, particularly during a pandemic when the topics are highly salient.

## Theoretical frameworks and background

Studies of science and health information acquisition usually focus on adults. The internet is currently the dominant source where adults acquire both scientific and health information, as well as general news [15,31]. In this paper we use two social psychological theories—JIT [15] and Archer's theory of science capital [14]—to understand how adolescents learn about science and health during a global pandemic.

People acquire information that is important to them, and the higher the salience of a topic to an individual, the more likely they will seek information on that subject [15,32]. Miller et al [15] describe this salience-driven model of information acquisition as the JIT. In the JIT news ecosystem, people get information as it becomes available from Twitter and other social media, (online) newspaper headlines, nightly newscasts, and personal social networks. Push notifications can appear at any time to alert people when new information is available. Although public libraries provide a physical place for people to seek information, personal electronic media makes news immediately available and therefore dominates information access [33].

Among people in the U.S., information acquisition behaviors for science topics differ from those for health [33]. Miller and colleagues suggest these two topics become relevant to individuals for different reasons; namely, interest in health information is often driven by utilitarian motives, and interest in science information often by curiosity [15]. Good health is

generally important to people, and they seek information about their own health and that of their families and friends. Science information acquisition is usually less personal—people may seek information to be better informed, for work, or because they are curious [15,33]. Miller and colleagues suggest that people who are driven to seek scientific information by curiosity are more thorough than those who seek health information for utilitarian reasons.

Youth vary in their access to different types of media compared to adults. Content may be explicitly monitored by parents to varying degrees within households based on youth age, access to technology, and social class. According to Coyne et al. [9], youth aged 8–18 who are from lower income families average almost two hours more of "entertainment" screen time per day compared to youth from higher income families [9]. Generally, in 2019, Common Sense Media found that tweens aged 8–12 spend just under five hours using non-school-related screen media per day and teens aged 13–18 use just under eight hours [34]. Since Common Sense Media's last census in 2015, the number of youth who watch videos online has doubled, from 24% to 56% among 8- to 12-year-olds, and from 34% to 69% among 13- to 18-year-olds [34]. Several factors influence how youth interact with and make meaning from their screen time. Youth media consumption can reflect implicit and explicit socialization by guardians as well as parental monitoring [9]. Youth might implicitly model guardians' behaviors or actively participate in family discussions about news and current events [10,35]. Edgerly et al. [7] found four media repertoires among a nationally representative sample of adolescents: news "avoidant" youth were approximately 52% of the sample. Another 15% were youth who accessed media through "curated-news-only." These youth encounter news media passively through curated sources as part of the algorithm in their social media (e.g., Instagram, Twitter, Facebook). Another 19% seek out news media primarily via legacy news organizations through the radio, internet, or television; and the fewest number, 14%, are "news omnivores" who seek out media from multiple different sources [7]. The data for the study conducted by Edgerly et al. [7] was collected in 2014. It does not assess the extent to which youth are accessing science and health information, nor does it ask about a specific topic such as COVID-19.

The deliberately misleading messaging on social media platforms [36] also complicated access to accurate information about COVID-19. Youth inexperience with news media could mean that they are especially vulnerable to the proliferation of "fake news" on social media and the internet [37]. Indeed, recent research on civic online reasoning found that 80% of middle-school youth believed that online native advertising (promotional ads formatted to look like news that can also show up in social media feeds) were real news stories [38]. Kiili and colleagues similarly found that a substantial proportion of adolescents were unable to critically assess online health-related information [39]. Even youth who access legacy media sources may encounter confusing or misleading messaging and may misplace trust in political leaders. One study conducted during the pandemic found that reliance on legacy media outlets and higher trust in governmental leaders in the U.S. and the United Kingdom (Donald Trump and Boris Johnson, respectively) was associated with lower health literacy related to COVID-19 among adolescents [40].

In addition to influencing youth media consumption, evidence suggests that parents and family influence youth interest in science [11,13,14]. Archer and colleagues developed a theory of science capital to illuminate how families cultivate science activities, identities, and future career aspirations associated with their social location (e.g. social class, race/ethnic, gender) [11,14]. Science capital comes in several forms, including books about science, being related to scientists, visiting science museums, watching science documentaries, and talking about science topics. Studies using the science capital framework show that low science capital is associated with developing lower science identities [11].

Access to science capital is unequal and distributed disproportionately along historic patterns of inclusion and exclusion in science based upon gender, race, ethnicity, ability, and location [16,41]. As with other types of capital, families that are more affluent can provide more sorts of science-related opportunities and experiences to their children than working class families [42]. Youth whose parents have a college education and many books in the home are also more likely to visit libraries, zoos, and museums and to engage in other informal science activities that contribute to science identities [13,16,43]. What appears to be an association between sources of and accuracy of COVID-19 knowledge could simply reflect not only fundamental science capital [14] but also higher science identity. Science identity is a self-concept as a science kind of person [44,45]. Because identities shape behaviors, youth with higher science identities are more likely to seek out sources of information about science topics [46,47]. Youth with higher science identities may also have higher levels of health media literacy, and may be more motivated to seek out science out of curiosity or for utility, including seeking out information about a novel topic such as a global viral pandemic.

Stories about the COVID-19 pandemic are not quite like the entertainment media tracked by Common Sense Media, nor are they exactly like the sources of health and science information that Miller et al. studied among adults. The COVID-19 pandemic has been covered heavily in the news media, and it also became increasingly politicized, especially in the U.S. [48]. It is not clear to what extent youth actively avoid media about news and politics [7]. It is unclear if youth avoid news and politics to the same extent today given that media use has changed since 2014 [34], and media consumption patterns may have changed even more during a global pandemic. Because COVID-19 is highly salient to youth, youth may access and interpret news differently than they did before the pandemic.

Research conducted during the pandemic suggests that youth in global North countries tended to seek science and health information from sources most familiar to them. Waselewski and colleagues identified legacy news media as the most common source for COVID-19 information for youth in the U.S. [40,49], however, youth also looked for information on social media, and they acquired information from friends and family. Norwegian youth surveyed in 2020 had a similar profile: they relied upon legacy media and friends and family as their primary sources of COVID-19 information but also reported using social media [50].

Regardless of how and where they search for information, how much young people know about the pandemic is unclear. Austrian adolescents surveyed during the pandemic generally demonstrated low levels of general knowledge about viruses and vaccination [51]. These studies [49–51] offset the relative paucity of research on how youth seek science and health information about the pandemic outside of what occurs in schools. Studies from before the COVID-19 pandemic that examined adolescents' understanding of viral diseases and epidemiology indicate generally low levels of knowledge [52]. The Next Generation Science Standards, the latest research-based content standards being implemented in many U.S. schools, do not include specific topics related to viruses, vaccines, or immunization [53]. Therefore, evidence from before COVID-19 suggests that youth had little prior knowledge about viruses and immune responses to form a basis for mental scaffolding to successfully add new information. Yet the high salience of the pandemic and associated restrictions, plus the ubiquity of news reports, could have supported rapid accumulation of COVID-19 specific knowledge.

To extend Miller's JIT model from adults to children, we must account for social and science capital. Why? Because so much of the media that youth consume and extracurricular experiences that they can access are influenced by the social class of their family [10,13,42]. Therefore, we must account for social location (parental education, # of books in the home, race/ethnicity, gender) [42]. We also need to assess science capital through what Archer [14] calls "science behaviors." Science behaviors include visiting libraries, science museums and

zoos and watching shows about science or nature. Even though youth may not have the autonomy to visit science museums or zoos without parental capital, most youth have choices about watching science shows and visiting libraries [16,54]. To account for youth agency we need to include several measures of curiosity about science that are relevant to science information acquisition in Miller's JIT theory [7,15]. Further, JIT theory suggests the need to measure youth perceptions of the utility of science for health decisions and for improving society, because salience of science is likely to matter for acquisition of accurate knowledge about COVID-19.

In summary, science capital theory suggests that youth with higher science capital will also have higher science identities and more positive attitudes about the utility of science for health and society. Higher science capital should be associated with having more types and a higher number of sources of science and health information. JIT theory suggests if youth perceive science as salient for health, they will acquire more information about topics such as COVID-19 [15]. We see value in integrating the theories to highlight the potential for science capital, science identity, or attitudes towards the utility of science for health and society to modify a JIT measure such as the number of sources of information and accuracy of COVID-19 knowledge. We test the utility of integrating science capital and JIT by modeling, with interaction terms, if the association of number of sources with the accuracy of knowledge depends upon science capital and/or attitudes such as the utility of science for health or society.

For the present study, we are interested in how youth learned about the COVID-19 pandemic and if type or number of sources is associated with higher accuracy of COVID-19 related knowledge. To explore these questions about COVID-19 knowledge, we surveyed youth in a moderately sized Midwestern U.S. school district and asked about their sources of information about COVID-19. We also embedded an experiment in the survey with random assignment to comic stories with scientific and health information about COVID-19. This study provides a quantitative analysis of the open-ended responses to the sources of knowledge and an assessment of the experiment. We also include a multiple regression analysis of the association of sources of knowledge with accuracy of knowledge, adjusted for science capital, science identity, attitudes about the utility of science for health and society, social location, experimental condition, and school learning type (in-person, remote, or hybrid).

## Methods

### Study design and sample

The data for the current study were part of a National Science Foundation Rapid Response Research project designed to educate youth about COVID-19 as quickly as possible in the summer of 2020. A large multidisciplinary team created free, weekly online comics starting mid-summer 2020. The C'RONA Pandemic Comics webpage [55] and book [56] built on a decade of expertise creating comics about the biology of viruses. The goal of the comics was to help adolescent readers understand the complexities of living through a viral pandemic by gaining accurate knowledge of the virus, the human immune system, connections between human and animal health, and historical and contemporary community responses to contagious diseases. In collaboration with virologists, social scientists, members of Native communities, and well-known comic artists, the team developed three comic stories—fictional narratives—that address fundamental issues in infectious disease and virology, including a story about a U.S. Tribal community response to the pandemic in historical context. For a detailed description of the C'RONA Pandemic Comics' creation and for an overview of the comics, see Diamond et al. [57].

To evaluate if youth gain knowledge through reading the comics and to discover sources of information that youth usually access, we conducted an online survey with an embedded experiment on the comics. Between December 2020 and February 2021, we sent electronic invitations to approximately 15,000 students enrolled in 5th through 9th grade in a mid-sized Midwestern urban school district, through a parental/guardian contact, to participate in an online survey study. All procedures and instruments were approved prior to use by the Institutional Review Board at the University of Nebraska-Lincoln (IRB Approval #: 20201120572EP) and the evaluation and assessment unit of the collaborating public school district. Parental consent and youth assent were obtained electronically. To increase participation, we sent three reminders to the children of the parents who elected to participate, as well as to youth if parents provided an email contact. Even with the reminders, only 280 youth participated (less than 2%). The low response rate was likely due both to the ongoing pandemic and the need for affirmative parental consent plus parents needing to provide access to youth (the school district would not allow us to contact youth directly). Although the low response rate reduces the statistical power, the results are nevertheless unique and contribute to our theory and understanding of youth information sources and knowledge during the COVID-19 pandemic. If we used a case-wise missing approach, we would have an analytic sample of 242. The question about how often youth visit public libraries had the most missing data from youth skipping the question (n = 22). For some questions youth could select "I don't know" as a valid value, but "I don't know" was not a response category for the question about visiting public libraries. To retain as many cases as possible we converted the missing values to "I don't know" and assigned the value of the mid-point of the scale to these cases, similar to the practice we describe below for variables that had "I don't know" as a valid value [58]. Sensitivity tests with and without these missing cases provided substantially similar results. The analytical sample therefore consists of 264 participants.

One purpose of the study was to evaluate if youth could learn accurate COVID-19 relevant knowledge from comics. There was one longer comic (10 pages) and two smaller comics (five pages) in the series. To create equal conditions, Comic I was separated into two comics that were five pages. Therefore, we had four randomized comic conditions that were five pages each. Although Comic I was 10 pages, the narrative did not appear as disjointed to youth due to formatting and headings that differentiated the first five pages from the second five pages. Additionally, the information presented to youth was different between the two five-page sections in Comic I, so we were able to formulate different knowledge-based questions across these two conditions. The four comic conditions were: C'RONA COMIX Pages 1–5, C'RONA COMIX Pages 6–10, C'RONA COMIX II Pages 1–5, TRIBAL C'RONA COMIX Pages 1–5.

Analytic information from the Qualtrics software program indicated that the five pages of comics took an average of 10 minutes to read, and the survey took approximately 20–30 minutes to complete. All of the current study participants, as well as all of the middle school youth in the study sampling frame, were given hard copies of the C'RONA Pandemic Comics books [56] after the data collection for the current study. An online survey with open-ended and closed-ended questions was used to gather the data for this study.

## Analytic strategy

The current study involved several steps, in part because we were simultaneously assessing the extent to which comics convey accurate information about COVID-19 during the pandemic and assessing existing sources of information and the need for more. The experiment provided subsets of youth with different kinds of information about COVID-19 and therefore had the potential to differentially raise the level of accurate knowledge. To account for the

experimental results in the focal analyses of youth-reported sources of information about COVID-19 and accuracy of COVID-19 knowledge, we had to account for the experiment. In addition, there were no existing scales to measure the knowledge provided in the comics, therefore we had to construct those measures. Finally, no list of sources of information about COVID-19 for youth existed at the time, therefore we used an open-ended question and constructed measures from the responses. The analyses therefore have three parts. First, we explain the items that make up the focal dependent variable, COVID-19 relevant knowledge. In the measures sections we describe how we developed the items, and in the results, we provide t-tests for whether youth gained knowledge from the comic they read, and then we provide descriptive statistics about youth accuracy on each of the items for the subscale on COVID-19 relevant knowledge. Second, we describe the process of converting open-ended responses to quantitative measures of the focal independent variables, kind and number of sources of information about COVID-19. Third, guided by theories of science capital and JIT, we used multivariate regression to model the association of sources of information about COVID-19 and accuracy of COVID-19 relevant knowledge, adjusted for control variables [10]. We also model, with interaction terms, if the focal association is modified by science capital, science identity, or attitudes toward the utility of science for health and society [15].

## Concepts and measures

**Dependent variable.** *COVID-19 relevant knowledge*. For all study participants, the dependent variable measures COVID-19 relevant knowledge. Because the study included randomization to one of four conditions, we first needed to adjust comic condition. Prior to the start of the survey, all participants were asked to read five pages of a comic story that contained accurate scientific information about COVID-19 protective health measures, the human immune system, bats and COVID-19, or the impact of COVID-19 on a Tribal community. The research team worked closely with scientists and content creators to develop 20 knowledge items about COVID-19, health preventative behaviors, viruses, vaccines, the immune system, animals and public health, and historical facts. To assess knowledge of factual information presented in each of the comic conditions, these 20 true/false statements were designed specifically to align with the content in each of the four comic treatments. The number of items per comic ranged from three to seven. The research team also developed five true/false items assessing more general COVID-19 and other health-related knowledge not specifically addressed in any comic set, which resulted in a total of 25 knowledge items in the survey. To avoid inadvertently spreading misinformation, all the factual statements included in the survey were true. A list of all 25 knowledge questions and those included in comic knowledge scale and COVID-19 relevant knowledge scale are available in S1 Appendix.

To a create a general measure of COVID-19 relevant knowledge for the multivariate analysis, we focused on a subset of 19 items specifically related to the COVID-19 pandemic and excluded factual items that were relevant to the comic story but unrelated to health or science. See details about each item that makes up the 19-item scale and youth accuracy on individual items in the results section.

## Focal independent variable

**Sources of information about the COVID-19 pandemic.** To understand the role of the comics within the context of existing sources of information about COVID-19, the survey included an open-ended question about where youth got information about the pandemic. We knew of no comprehensive list of sources of information about COVID-19 for youth, so we chose to use an open-ended question with no indication of any type of source (e.g., media,

family, friends). The survey asked youth, "What are your main sources of information about the COVID-19 pandemic?" Almost all youth (96%) provided some written response to this query, although a few (7%) simply listed pandemic knowledge or replied without answering the question. Two researchers iteratively created exhaustive thematic codes based on all substantive responses. We identified four categories of codes: Media, Family/Friends, School, and Experts. Youth varied in specificity of their sources. For example, many youth simply said "news" while others listed several media types (e.g., social media, internet, TV) or several news sources (e.g., CBS, CNN, Fox). While we wanted to assess whether the type of source (Media, School, Family/Friends, Experts) mattered for COVID-19 knowledge, we also wanted to quantify distinct sources of information youth listed.

We created a count of COVID-19 sources of information from the open-ended question to assess the relationship between the number of sources and COVID-19 relevant knowledge. A dataset including all open-ended responses and codes is publicly available here [provide DOI to publicly available data]. See more information about the frequency of codes and the number of information sources in the results section.

## Other focal independent variables

**Science capital.** In addition to measures of social location, we measure what Archer et al. [14] calls "science behaviors" to assess science capital. Youth were asked how often they visit libraries, science museums, zoos, and how often they watch shows about science or nature. The response categories for zoos, science museums, and watching shows about science and nature are "never" (0), "once in a while" (1), "sometimes" (2), and "often" (3). For how often youth visit public libraries we imputed 22 cases to the mid-point of the scale similar to "I don't know" measures on the items about the utility for science for health and society. The response categories are "never" (0), "once in a while" (1), "I don't know" (2), "sometimes" (3), and "often" (4).

**Science identity.** To better model the unique association of sources of information and accurate knowledge, we include a measure of science identity. Similar to Hill et al. [59] we measure science identity with a single item variable that asked youth how much they think they are a "science kind of person." Response categories ranged from "not at all" (0), "a little" (1), "somewhat" (2), to "totally" (3).

**Utility of science for health and society.** Miller's theory of JIT indicates that some people may be more motivated to seek out health and science information than others, and motivations may differ depending on whether the information is about science or health [15]. To assess this, we asked youth, "How much does science help you make decisions about your body?" and "How much does science help people?". Youth were allowed to respond "I don't know" (12.1% and 3% respectively. We recoded those students at the mid-point. Response categories ranged from "not at all" (0), "a little" (1), "I don't know" (2) "somewhat" (3), to "a lot" (4).

**Social location and learning environment.** We also included measures to account for SES (# of books in the home, whether a parent/guardian graduated college) because of likely correlations with science behaviors, science identity, and science attitudes. Because of historical discrimination and implicit biases, girls and students of color do not have the same access to science identities as youth who identify as male or who are white [41,59]. Additionally, in adult populations, Miller found that women are more likely to seek out health information for utility reasons while men are more likely to seek out science information out of curiosity [15,33]. We include gender and race/ethnicity as control variables for these reasons. Additionally, we control for grade level and comic condition.

## Results

### Comics as sources of information about COVID-19

Can comics effectively convey scientific knowledge about COVID-19 to youth? To answer this question we used t-tests to compare whether youth who received each comic condition were more accurate (percent correct) on the items that aligned with their comic than youth who did not receive the comic pertaining to those items. The first comic (comic condition one) contained information that was often in the news such as the importance of physical distancing and masking to help stop the spread of COVID-19. Yet youth who were assigned to read this comic had, on average, 7% higher scores on the relevant survey items than youth who did not read this comic (p < .05). The other three comic conditions contained information that was less common in the news. The average differences between those who did or did not read a comic with information on the remaining topics ranged from 13% higher for condition three (about bats and One Health) to 24% higher for condition four (about Tribal experiences during the pandemic) to 27% higher for condition two (about the immune system) (p < .01) (results not shown in table).

After assessing whether youth gained knowledge from comics on the 20 comic-aligned knowledge questions, we then look at general accuracy using the 19-item COVID-19 relevant knowledge scale. For a detailed description of the 25 knowledge items and how they aligned with the comics and whether they were included in the COVID-19 knowledge scale, see S1 Appendix. Fig 1 shows the 19 true/false items that comprise the measure of COVID-19 relevant knowledge organized by different science/health content areas.

Overall, the average percentage correct (81%) was relatively high. The first five statements were about behaviors to contain the pandemic and the percentage of youth with accurate scores, ranged from 88% to 97%. Similarly, scores ranged from 85% to 94% of youth with accurate answers for the next four statements that were specific to the COVID-19 virus

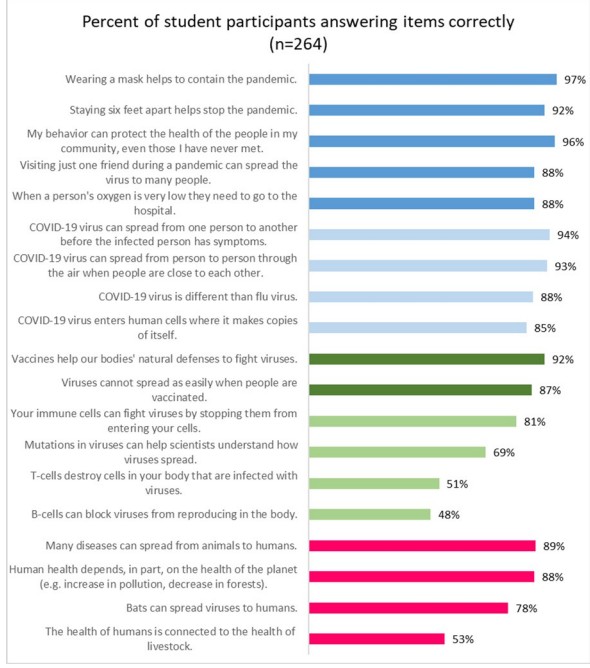

**Fig 1. Percent of accurate responses to questions about COVID-19 relevant knowledge.**

transmission and traits. Many participants also had correct answers to the two questions in the third set of items that were about vaccines, with 87% correct about viruses not spreading as easily when people are vaccinated and 92% correct that vaccines help our bodies fight viruses. Fewer youth had accurate knowledge of the immune system, with agreement to statements about B-cells (48%) and T-cells (51%) quite low, and the more general questions about viral mutations (69%) and immune cells (81%) somewhat higher. Only about half of the participants (53%) knew that the health of humans is connected to the health of livestock, yet more had accurate knowledge (78% to 89%) on other statements about how human health is connected to other living animals and our shared environments (i.e. One Health [60]).

## Youth-reported sources of information about COVID-19

What sources of information did youth have about COVID-19 during the pandemic? To find out we summarized the responses to the open-ended question about usual sources of information. Table 1 shows the distribution of responses in four broad categories: (1) *Media* (67%); (2) *Family, friends, and acquaintances* (47%); (3) *School* (39%); and (4) *Experts* (17%). Table 1 also provides the proportion of youth who listed specific types of sources that fall within the four broad categories. *Media* included any mention of news, internet, social media, radio, books, TV, plus the C'RONA Pandemic Comics that were part of the study. When students named specific news organizations (e.g., CNN, MSNBC), we counted each source. We also separated legacy media mentions from social media or internet mentions (not shown in the table). *Family, friends, and acquaintances* included any parent or guardian (sometimes more than one),

**Table 1. Sources of information about the COVID-19 pandemic (N = 264).**

| What are your main sources of information about the COVID-19 pandemic? (n = 264) | | |
|---|---|---|
| **Category** | **Specific Code** | **% mentioned** |
| Media (67%) | News | 53% |
| | Internet/online | 17% |
| | TV | 7% |
| | C'RONA Pandemic Comics | 5% |
| | Social media | 4% |
| | Radio | 2% |
| | Books | 2% |
| | YouTube | 2% |
| Family, friends and acquaintances (47%) | Parent(s) | 39% |
| | Family | 5% |
| | Other people | 5% |
| | Friends/peers | 5% |
| | Grandparents | 2% |
| School (39%) | School | 28% |
| | Teacher(s) | 9% |
| | Science class | 8% |
| | COVID-19 lesson | 2% |
| Experts (17%) | Doctors/ Healthcare workers | 11% |
| | CDC/ WHO/ local health department | 7% |
| | Scientists/ researchers | 3% |
| | Specific experts | 2% |

\* Note: Because youth could respond with more than one source, percentages include any mention and equal more than 100%.

grandparents, other family, other people more broadly, and friends/peers. School included cases when students just said "school" broadly, if they mentioned a specific teacher, if they specifically mentioned a science class, or if they said they had a lesson or did a project about COVID-19 in school. *Experts* included youth reports that they learned about COVID-19 from a healthcare provider, a government agency, researchers, or scientists, or if they listed any other expert sources (e.g., Dr. Fauci). If youth reported that their parent was a healthcare provider or expert, we counted that as two sources of information as they noted their parents' dual roles. In Table 1 we arranged the responses so that the most common sources are listed first. The cumulative percentages can exceed 100% because youth could list numerous sources for their main sources of information about the COVID-19 pandemic. A detailed dataset with youth open-ended responses and codes is available here [DOI to publicly available dataset].

Students varied in how much explanation they gave about the sources that they used for information about COVID-19. Just under a third of youth simply said, "The news," but others provided specific details, for example:

- "The New York Times and Anthony Fauci."

- "CBS This Morning, CNN, CDC, and the Mayo Clinic, But I mainly watch CBS the most."

- "www.foxnews.com"

- "The news, but mostly CBS."

- "cdc.gov fox news cnn news cnn 10"

- "CNN News, Channel 8 eyewitness news"

- "My parents and NPR/podcasts which my mom listens to in the car to and from school."

- "News sources such as CNN or any other news network"

- "Accurate news channels"

Some of the youth emphasized that they had multiple ways of accessing information, and about 5% mentioned the comics:

- "The comics, The news, My parents, and the radio."

- "My parents and what I have heard from health care professionals online, on TV, and occasionally in person."

- "Various articles, CDC, these comics, and a video I watched in a class"

- "I guess the news and school also my doctors plus I search it up if I want to find out something about the virus."

Others noted their access to parents or other adults who served as information sources:

- "My mother and father were front line workers during the pandemic of Covid—19. They were my main source of information."

- "my science teacher does covid fridays where he gives us a few websites and a few questions and we cover different topics. we've gone over things like what the virus is, other viruses and pandemics, the effectiveness of masks, social distancing, washing your hands, and sanitizing your hands/surfaces. that's where i've learned most of the stuff i know about covid-19."

- "My grandpa, who is a doctor, knows a lot about the virus, so he helps my sister and I understand how it works. We call him when we are sick or hurt so that he can help us find the best

way to treat the injury or sickness. My other source is the internet, where I read articles about the virus and how I can protect myself from it."

- "School, my dad who works in the health department and the cdc. Also the news like 1011 or today show."

Demonstrating awareness of variation in the quality of sources of news, some respondents commented on the reliability or validity of their information sources:

- "When i go to school in the morning i hear facts about the covid-19 things on the radio, I also ask my mom if i have questions about it. If i hear something from a Source i will check other sources to see if they agree with the information i found on that particular source. I try to fact check as much as i can just in case i am wrong."

- "Scientific research from doctors, or people that study COVID and they are very reliable."

- "Any credible source. We had a unit on COVID-19 in school that also boosted my knowledge."

We anticipated that many youth would mention the internet or social media as a source of information, but only a few did (4%), as in the following responses:

- "Tik tok (The anime) cells at work that help me learn about what cells do what when there's a virus or disease or what each cells Job is. And my mom that makes sure that every think I know about this is true (or she thinks) so I can stay safe and not get sick."

- "My parents, Reddit"

- "Youtube"

- "my teachers memes youtube"

- "Websites, YouTube videos, random articles and school"

Some youth reported that they got information through members of their social network. For example, one participant indicated that, in addition to the news, "sometimes the word about something related to COVID-19 gets around school." Others gave the following responses:

- "Family, school, friends"

- "News, Signs or Rumors"

- "People"

- "The internet and people around me"

Result from the open-ended question about media use support previous findings on the central role of mainstream or "legacy" media for youth during a pandemic. More than half (55.7%) of the youth got information about COVID-19 from legacy or traditional news sources, compared to about 90% among adults [61]. While youth named similar sources to adults, they also relied heavily on the adults in their lives (e.g., parents, caregivers, grandparents, teachers) as their main sources of information. A total of 39% mentioned that they learned about COVID-19 from school, including pandemic-specific lessons from teachers and school-based media. At least nine students explicitly mentioned CNN 10, an on-demand digital news show composed of 10-minute segments created by CNN for students aged 6–12 that is commonly used in class by teachers. We next turn to analyses in which we explore how the

type of sources, number of sources, and depth of description (i.e. number of words) are associated with accuracy of COVID-19 relevant knowledge.

## Univariate and bivariate results

Table 2 provides univariate descriptive statistics for the study variables. As noted above, on average had COVID-19 relevant knowledge scores of 81% correct (SD = .16). Youth wrote on average seven words (M = 6.85, SD = 9.12) in response to the open-ended prompt; the range of the number of words was large, from zero to 71. The average number of sources about COVID-19 was just over two (M = 2.11, SD = 1.23); 97% of youth reported between zero and four sources; the highest number was six. In response to the questions about informal science activities, youth reported visiting public libraries most often, with an average score close to the value for "sometimes" (M = 2.02, SD = 1.43), followed by watching shows about nature and science (M = 1.72, SD = .91), visiting a zoo (M = 1.69, SD = .80), and visiting a science museum, which had an average close to the value for "once in a while" (M = 1.13, SD = .75). Youth reported that science helps them make decisions about their body on average between "a little" and "somewhat" (M = 2.83, SD = 1.19), and that science helps people between "somewhat" and "a lot" (M = 3.77, SD = .63). Compared to other studies [59], youth in the sample had fairly high science identities: between "somewhat" and "a lot" (M = 1.84, SD = .88).

More youth in lower grades (23% in 5th, 29% in 6th, 24% in 7th grade) participated than youth in higher grades (16% in 8th and 8.3% in 9th grade) (results not shown in table). The sample had fewer students of color (21%) than the district as a whole (35.8% students of color). Additionally, the sample had a slightly higher percentage of boys (53%) than the district (51%).

**Table 2. Descriptive statistics for sample variables from youth (n = 264).**

| | Mean/ Proportion | SD | Min. | Max |
|---|---|---|---|---|
| COVID-19 Relevant Knowledge (Proportion Correct of 19 items) | .81 | .16 | 0 | 1 |
| What are your main sources of information about the COVID-19 pandemic? | | | | |
| word count | 6.85 | 9.12 | 0 | 71 |
| # of sources | 2.11 | 1.23 | 0 | 6 |
| Informal Science Activities | | | | |
| How often do you visit the public library? | 2.02 | 1.43 | 0 | 4 |
| How often do you visit a science museum? | 1.13 | .75 | 0 | 3 |
| How often do you visit a zoo? | 1.69 | .80 | 0 | 3 |
| How often do you watch shows about science or nature? | 1.72 | .91 | 0 | 3 |
| Utility of Science & Identity | | | | |
| How much does science help you make decisions that affect your body? | 2.83 | 1.19 | 0 | 4 |
| How much, if at all, does science help people? | 3.77 | .63 | 0 | 4 |
| How much do you think you are a science kind of person? | 1.84 | .88 | 0 | 3 |
| Social Location | | | | |
| Male | .53 | | 0 | 1 |
| Non-minority | .79 | | 0 | 1 |
| >100 Books | .67 | | 0 | 1 |
| Parent attended college | .63 | | 0 | 1 |
| Learning Environment | | | 0 | 1 |
| Remote | .22 | | 0 | 1 |
| In-person | .64 | | 0 | 1 |
| Hybrid | .14 | | 0 | 1 |

Approximately 67% of the sample reported more than 100 books in the home, and 63% reported at least one parent or guardian had attended college. The findings provide a snapshot of how students obtain information and what they know of the COVID-19 pandemic during a historic time, yet we cannot claim to represent the whole district or all youth in the U.S.

The same historic conditions that led us to study youth in 2021 also led to many challenges for students and schools. In our sample, 64% of youth attended school mostly in-person, 22% attended mostly online, and 14% reported attending a hybrid model. All youth who were in classrooms had peers joining via Zoom, and the majority of students who attended in-person wore masks. The findings provide a snapshot of how students obtain information and what they know of the COVID-19 pandemic during a historic time, yet we cannot claim to represent the whole district or all youth in the U.S.

We also conducted bivariate correlations and found that increases in the number of sources was associated with higher accuracy of COVID-19 knowledge (Pearson's r = .23, p < .001). There was a weaker and non-statistically significant association of word count and COVID-19 knowledge (Pearson's r = .118, p = .056). We created dummy variables for each source type and used t-tests to assess if any type of source was associated with more accurate COVID-19 relevant knowledge compared to the combination of the other types of sources. We also did a sensitivity test in which we broke down "Media" into legacy media versus online/social media similar to recent studies [40,49]. We found that youth who mentioned any legacy media source (TV, radio, books, "news", or a specific news source) had higher average higher scores on the COVID-19 knowledge scale compared to youth who did not mention a legacy source (83.8% vs. 78.2%, p < .01). The remaining COVID-19 mean knowledge comparisons (i.e. youth who mentioned the internet or social media, school, family, friends or acquaintances, or experts as sources versus those who did not) failed to reach statistical significance.

## Multivariate results

Are sources of information associated with accurate COVID-19 knowledge? To answer this question, we used ordinary least squares multiple regression and added variables in four models (see Table 3). All ordinal and continuous independent variables were mean-centered to reduce multicollinearity and to simplify the interpretation of the intercept. The reference categories for the categorical variables were: assigned to the comic story about a U.S. Tribal community response to the pandemic, girls, youth of color (minority), those with >100 books in the home, those whose parents/guardians have attended college, those in fifth grade, and those who have the mean on all other items. We initially also included type of media source (legacy compared to all others) in the model, but due to high multicollinearity with the number of sources measure, it was not significant and therefore we did not include it in the final analyses reported in Table 3.

In Model 1 of Table 3, we show the association of the number of COVID-19 information sources with COVID-19 relevant Knowledge as the outcome, adjusted for the control variables. The coefficients for the control variables are not listed in Table 3; none of them had statistically significant associations with accuracy of COVID-19 relevant knowledge. The association between number of sources of information and COVID-19 knowledge was positive and moderate (Beta = .229, p < .01). The unstandardized coefficient is about a fifth of the standard deviation in the outcome (SD = .16/B = .03). The control variables plus sources of knowledge accounted for 3.8% of the variance in COVID-19 knowledge (R-squared = .038).

Model 2 in Table 3 added indicators of science behaviors/science capital. The positive association between the number of COVID-19 sources and accuracy of COVID-19 knowledge remained positive and statistically significant in Model 2, although it is smaller (Beta = .176, p

**Table 3. Multiple regression of COVID-19 relevant knowledge on information sources.**

| Table 3. Multiple Regression of COVID-19, Viruses, and Vaccines Knowledge by (n = 264) | | | | | | | | | | | | | | | | | | |
|---|---|---|---|---|---|---|---|---|---|---|---|---|---|---|---|---|---|---|
| | Model 1 | | | | Model 2 | | | | Model 3 | | | | Model 4 | | | | | |
| | B | S.E. | Beta | Sig. | B | S.E. | Beta | Sig. | B | S.E. | Beta | Sig. | B | S.E. | Beta | Sig. | | |
| # of COVID-19 information sources | .029 | .008 | .229 | ** | .023 | .008 | .176 | ** | .016 | .007 | .128 | * | .017 | .007 | .134 | * | | |
| How often do you. . . | | | | | | | | | | | | | | | | | | |
| visit a public library? | | | | | .022 | .007 | .201 | ** | .018 | .006 | .167 | ** | .019 | .006 | .175 | * | | |
| visit a science museum? | | | | | .028 | .014 | .136 | * | .007 | .013 | .032 | | .007 | .013 | .036 | | | |
| visit a zoo? | | | | | -.013 | .013 | -.066 | | -.110 | .012 | -.054 | | -.012 | .012 | -.056 | | | |
| watch shows about science or nature? | | | | | .028 | .011 | .161 | * | .003 | .011 | .017 | | .004 | .011 | .025 | | | |
| How much do you think you are a science kind of person? | | | | | | | | | .037 | .012 | .205 | ** | .030 | .012 | .170 | ** | | |
| How much does science help you make decisions that affect your body? | | | | | | | | | .028 | .008 | .215 | *** | .026 | .008 | .198 | * | | |
| How much, if at all, does science help people? | | | | | | | | | .037 | .015 | .150 | ** | .037 | .012 | .170 | * | | |
| # of COVID-19 information sources X How much does science help you make decisions about your body? | | | | | | | | | | | | | -.021 | .006 | -.196 | *** | | |
| Constant | .789 | .037 | | | .793 | .036 | | | .788 | .033 | | | .796 | .032 | | | | |
| Adjusted R-squared | | | .038 | * | | | .127 | *** | | | .264 | *** | | | .301 | *** | | |

All models control for comic book condition, gender, race, #books in the home, parent/guardian college, grade level, and learner type (remote, hybrid, in-person). In model 2, boys had significantly higher knowledge than girls/other on the dependent variable. There were no other significant associations among control variables.

* $p < .05$,

** $p < .01$,

***$p < .001$.

$< .05$). Among science behaviors, frequency of visiting public libraries (Beta = .201, p < .01), watching science and nature shows (Beta = .161, p < .05), and visiting science museums (Beta = .136, p < .05) were associated with more accurate COVID-19 knowledge. Visiting a zoo is not associated with COVID-19 knowledge. It is unlikely that visiting the library itself increases knowledge of COVID 19; instead, it is more likely that youth who go to libraries and have more sources of knowledge or retain more of the information that they obtain. The R-square increased from .037 in Model 1 to .127 in Model 2, indicating a substantial and significant increase in explained variance (p < .001).

We add indicators of science identity (Beta = .205, p < .01) and perceptions of the utility of science for health (B = .215, p < .001) and society (B = .150, p < .01) in Model 3. All three measures, even controlling for the other variables, had significant and positive associations with COVID-19 knowledge. Yet two of the science behaviors (visiting a science museum and watching shows about science or nature) are no longer significant once the science identity and utility for health and society measures were included. Consistent with Archer's theory of science capital [14], we interpret this change in the coefficients as indicating that the relationship between watching shows about nature and science and visiting science museums and COVID-19 relevant knowledge was mediated by science identity (separate analysis not shown). Frequency of visiting public libraries continues to have a positive and statistically significant association with accuracy of COVID-19 knowledge in Model 3. In this model the indicator of number of sources was still positive and statistically significant (Beta = .128, P < .01) but about half the size that it was in the original model, suggesting that some of the association reflects a general orientation towards science and learning and is less about having multiple sources of information, consistent with JIT theory [15]. In this third model the R-square is nearly double the prior model, indicating that general science identity and perceptions of the

utility of science for health and society explain the variance in COVID-19 knowledge (R = .264, p < .001).

Science capital theory suggests that more sources of information (more science capital) will contribute to more science literacy, in this case accurate knowledge of COVID-19 [14]. Yet Miller and colleagues' [15] JIT theory suggests that the effectiveness of more sources will depend upon how salient and relevant people see health and science information for their lives. We therefore assessed if the focal association between number of sources and accuracy of knowledge was modified by science identity or the utility of science for health and/or society. We found that only the interaction of number of sources and the utility of science for health was significant. Model 4 shows that the attitudes about the utility of science for health modified the association between the number of sources of information and accuracy of COVID-19 knowledge. Adding the interaction in Model 4 also increased the explained variance by 5% (R-square = .301).

Fig 2 shows a plot of the interaction of number of sources by utility of science for health (i.e., "science helps me make decisions about my body") from Model 4. We restricted the range of the number of sources of information to zero to four because most participants (97%) had values within this range. We estimated the predicted proportion correct COVID-19 relevant knowledge based upon one standard deviation below and above the mean for utility of science for health. For youth with low scores on utility of science for health, each additional source of information is associated with an increase in accurate COVID-19 relevant knowledge by three percentage, or 12 points total between zero (70% correct) and four sources (82% correct). For those with higher than average utility of science for health attitudes, there was very little change in accuracy on COVID-19 relevant knowledge. Therefore the association of the number of sources and accuracy of COVID-19 relevant knowledge is stronger for those with lower utility of science for health attitudes.

## Discussion and conclusion

The COVID-19 pandemic was unprecedented to the extent that information (and misinformation) about health and science was seemingly everywhere, and the pandemic was highly salient in most people's lives, especially youth [2,36]. Youth were asked to adapt to many changes in

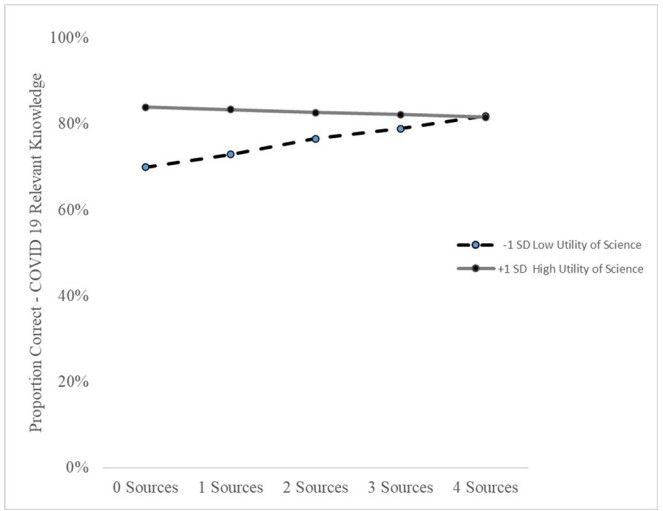

**Fig 2. Proportion accurate COVID-19 relevant knowledge by number of sources and attitude about the utility of science for health.**

schooling and activities based on information scientists and health officials were learning about the virus over time, through a filter of news media and politics [2,62]. As difficult as the pandemic was for so many people, the historic event also created a unique chance for researchers to explore how young people learn and adapt in a global health crisis [27,63]. Studying a health topic while it is relevant to most people is important because specific science and health information generally has salience only to small subgroups of people at any given time. It is rare that a science and health topic is covered in such depth and so often in the news, and information that is accessible, accurate, and tailored to engage youth can be difficult to find, particularly for youth with low science capital, science identity, or those who may avoid news media that is politicized [2,7,25,64,65]. Understanding youth sources of information about COVID-19 during the pandemic and their relevance for accurate knowledge provides insights for science educators and public health experts who often count on youth to follow protective health measures [50]. Identifying gaps in accurate information sources or that do not reach youth with low science capital can guide efforts to provide information that is accessible, accurate, engaging and relevant to the lived experiences of youth from many backgrounds [66].

Decades of research on youth science engagement, persistence, identities, and career aspirations [47] have provided relatively little information on how young people acquire emerging and new science and health information. While youth may learn about science and health from different sources than adults, they may understand information differently [20,67]. They also are more influenced by parents and their family context than adults [9,10]. They may have less autonomy over exploring their own interests due to differences between families in science capital [42], but they also may also have funds of knowledge that can help tie science to their lived experience [66]. Existing perspectives for studying youth and science provide a good starting point to conceptualize factors relevant to sources of information [7,9]. Archer and colleagues proposed a science capital framework grounded in the work of Bourdieu and others to conceptualize the components that contribute to youth science engagement and persistence [11,14]. Others focus on how youth navigate emerging media sources compared to conventional news media as a way to understand repertoires of news consumption [7] or how science-interested youth get information and make sense of COVID-19 [40].

The C'RONA Pandemic Comics were designed as a rapid dissemination of accurate information about COVID-19 through an entertaining medium. Because of the need to act quickly, we simultaneously provided accurate information and collected data on sources of information about COVID-19. The comics were an effective mechanism for conveying accurate information about COVID-19, particularly on topics that youth were less familiar with. Based upon comparisons between those assigned to specific comics or not, we discovered that more students had accurate knowledge of behaviors that could slow the spread of COVID-19 and less knowledge of how the immune system reacts to viruses, the role of animals in viral evolution and transmission, and tribal histories and experiences with viruses.

Our findings suggest that youth absorb accurate information from reading the comics: youth had higher accuracy on the topics covered in the comics that they read than on the topics covered by comics that they did not read. Spiegel and colleagues found that when almost all ninth-grade youth in a school district read comics or essays, those assigned comics were more likely to want to read similar materials, particularly those with lower science identity [32].

Prior studies of sources among adults compares legacy and online media [40,49]. Therefore we also compared level of accurate COVID-19 knowledge for those who listed legacy or online media and found that, at the bivariate level, those who reported legacy media did have higher accuracy than online, but the association was not significant when we included total number of sources. Therefore, it is possible that what seems to be a "legacy media" effect could simply be a "high number of sources" effect. Our findings suggest that youth absorb accurate

information from reading the comics: youth had higher accuracy on the topics covered in the comics that they read than on the topics covered by comics that they did not read.

The closing of schools; isolation from friends; cessation of sports, theatre, music, dance, and other enrichment activities; and concerns about the health of people in their lives (parents, grandparents, teachers) made the COVID-19 pandemic highly relevant to most youth [2–4,62,68]. Mainstream news media rarely "speak" directly to youth. Some outlets, such as NPR, involve youth reporters and feature youth voices while many teachers may utilize CNN 10 (student news) [69], yet only a few participants in the current study listed the 10 minute in-school news stories by name. Youth may encounter information about health and science from a variety of sources, and our study indicates that the more sources youth have, the more knowledgeable they are. Therefore, during a time when for most people "just in time" information is highly salient, those with more science capital will have more accurate knowledge. We hope that future research will discover if youth with more accurate knowledge also engage in more prevention behaviors.

Prior research indicates the majority of youth are news avoidant and many lack interest in mainstream media geared towards adults [7]. Our team identified that a sample of youth in a U.S. Midwestern school district obtained information about COVID-19 from many of the same sources used by adults, something that supports the findings of other studies conducted during this pandemic [25]. Although 5% of youth reported acquiring information from the comics we provided, they did not otherwise cite comics as a source. They generally relied upon legacy media, the internet, parents, school, and experts. The idea that many youth primarily learn from curated algorithms from social media [7] was not supported by the present study.

The youth generally had accurate COVID-19 relevant knowledge. Although parents and youth who were more interested in the topic, or who had higher science capital, may have been more likely to choose to participate in and complete the study, level of accuracy of COVID-19 relevant knowledge still varied in the sample. Youth with high science capital, high science identity, and more positive attitudes about the utility of science for health and society had more accurate knowledge. Yet even accounting for these general measures, we also found that increases in the number of sources about COVID-19 was associated with accurate COVID-19 relevant knowledge. Knowledge did not, however, vary by gender, race/ethnicity, social capital, comic condition, or grade level. For many science and health topics, we might expect that older youth would have a better understanding than younger youth; however, we found no difference in COVID-19 relevant knowledge by grade level. It is important to note that youth received accurate science and health information under each comic condtion, however. Indeed, in the internet era, information about a novel virus and the resulting pandemic spread rapidly from scientists to the public, to both youth and adults. Therefore, our findings demonstrate the importance of providing accessible and accurate science information about an emerging health crisis to all.

Indeed, during a time of dramatic global change, it is possible that rather than having less access to, or less accurate, knowledge about science and health information than adults, youth may have an advantage if schools are including up-to-date information or access to mainstream news sources in health and science classes. Providing accurate science about salient and relevant emergent health topics in schools alongside curriculum aimed at improving science, health, and media literacy may be critical to combating the rampant spread of disinformation in the future [18,70]. Some youth act as reservoirs of information for their families if they share science and health information they learn from school [71]. The dissemination of information is often portrayed as a one-way process from adults to youth. Decades of evidence, however, show that the process is non-linear and more closely resembles a network of transmission that varies with topic, family dynamics, social and cultural histories [33,34].

The current study has several limitations. Larger studies of youth have found an association between source or type of information and accuracy of COVID-19 knowledge [40]. The lack of an association of type of source and accurate of knowledge in the multiple regression results could reflect a lack of statistical power or that type of source serves as a proxy for number of sources. The bivariate positive and significant association of type of source is similar to results in larger studies with youth and adults [30,40]. To inform action for future health crises it will be important to determine if one legacy source can be as useful as multiple non-legacy sources or multiple sources that include legacy sources. Therefore we recommend future resource with youth that includes a set of possible sources now that we have a better sense of what youth are likely to list. A more detailed question with a list of possible sources of information, with a larger sample, will add useful information to clarify how best to measure a concept that is important to both social capital and JIT theory–sources of information.

We did not measure if more accurate knowledge is associated with behaviors aimed at preventing the spread of COVID-19 (e.g., wearing a mask, social distancing, getting vaccinated). One of the core assumptions for significance testing with multiple regression is that the data were collected using random sampling. In the current study, we sought participation from all public-school students in grades five through nine in one school district. Unfortunately, only a very small fraction of youth participated. Those who did participate over-represented some groups relative to representation in the district, thus limiting the overall generalizability of the findings. More research on the validity and reliability of the measure of COVID-19 relevant knowledge will also be essential. Members of our team created the items to reflect the knowledge conveyed in the C'RONA Pandemic Comics. The items cover many topics that are useful for people to know about COVID-19 as they navigate the implications, yet a more comprehensive list could be explored, particularly as new knowledge emerges.

Many youth described learning information from family members; it would have been useful to ask youth if they share information about COVID-19 with others. For example, if middle school science classes included skills for assessing the rigor or veracity of news sources and basic information about viruses, could youth help inform their families about how to assess the reliability of claims? Similarly, we did not receive many details about the informal flow of COVID-19 information through peer contacts. Future research needs to explore peer-to-peer spread of information during a global health crisis because it is an important and poorly understood element of information acquisition during a pandemic. Even though some students had less physical contact with friends and peers, many likely texted, talked on their phones or interacted with peers online.

## Supporting information

**S1 Appendix.**
(TIF)

## Acknowledgments

The authors would like to thank Grace Kelly, who contributed to this paper by helping to edit the paper, and by reviewing the graphs, figures, and qualitative coding during the revision process.

## Author Contributions

**Conceptualization:** Patricia Wonch Hill, Judy Diamond, Amy N. Spiegel, Julia McQuillan.

**Data curation:** Patricia Wonch Hill, Judy Diamond, Julia McQuillan.

**Formal analysis:** Patricia Wonch Hill, Judy Diamond, Julia McQuillan.

**Funding acquisition:** Patricia Wonch Hill, Judy Diamond, Elizabeth VanWormer, Julia McQuillan.

**Investigation:** Patricia Wonch Hill, Judy Diamond, Amy N. Spiegel, Elizabeth VanWormer, Julia McQuillan.

**Methodology:** Patricia Wonch Hill, Judy Diamond, Amy N. Spiegel, Julia McQuillan.

**Project administration:** Patricia Wonch Hill, Judy Diamond, Meghan Leadabrand.

**Supervision:** Judy Diamond, Julia McQuillan.

**Validation:** Patricia Wonch Hill.

**Visualization:** Patricia Wonch Hill, Judy Diamond, Amy N. Spiegel, Meghan Leadabrand.

**Writing – original draft:** Patricia Wonch Hill, Judy Diamond, Amy N. Spiegel, Elizabeth Van-Wormer, Meghan Leadabrand, Julia McQuillan.

**Writing – review & editing:** Patricia Wonch Hill, Judy Diamond, Amy N. Spiegel, Elizabeth VanWormer, Meghan Leadabrand, Julia McQuillan.

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
