## [Decision Letter · Decision Letter 0]

11 Oct 2022

PONE-D-22-11112Accuracy of COVID-19 relevant knowledge among youth: Do sources of information matter?PLOS ONE

Dear Dr. Hill,

Thank you for submitting your manuscript to PLOS ONE. After careful consideration, we feel that it has merit but does not fully meet PLOS ONE’s publication criteria as it currently stands. Therefore, we invite you to submit a revised version of the manuscript that addresses the points raised during the review process.

While the authors are welcome to address any of the reviewers' comments, the following are required changes:

Per reviewer 1: Please address this review’s comments regarding the abstract.

Per Reviewers 1 and 2: Please review the background and conclusions sections to ensure that the information provided is cited appropriately.

Per Reviewer 2: Please provide some context for why comics were chosen and any background, if applicable, from other studies that used comics to inform youth on science topics.

Per Reviewer1: Please provide information on how survey questions were developed in the methods section.

Per Reviewer 2: Please clarify the number of comic stories used in the study

Per Reviewer1: Please provide additional information about the coding process for qualitative data in the methods section.

Per Reviewer 1: Please revise the presentation of “qualitative” results. For example, beginning on line 339 there is a presentation of results in response to which sources youth gain scientific knowledge from, but this is a listing of news sources that could be treated as quantitative data as there is little qualitative information to be gained from it. This is also applicable to the data presented starting on line 432 and online 450. Other responses provide better qualitative information but that section needs to be revised to provide context about the data rather than just listing the quotes.

Per Reviewers 1 and 2: Please modify the results section to link the results to the aims of the study and to clarify the findings

We look forward to receiving your revised manuscript.

Kind regards,

Cindy Prins

Academic Editor

PLOS ONE

2. “Parts of this publication were also supported by the National Institute of General Medical Sciences at the National Institutes of Health under award R25GM129836 (JM, PWH) (nihsepa.org). Any opinions, findings, or conclusions expressed in this material are those of the authors and do not necessarily reflect the views of the National Science Foundation or the National Institutes of Health.

Reviewers' comments:

Reviewer's Responses to Questions

**Comments to the Author**

1. Is the manuscript technically sound, and do the data support the conclusions?

Reviewer #1: Yes

Reviewer #2: Partly

2. Has the statistical analysis been performed appropriately and rigorously? 

Reviewer #1: Yes

Reviewer #2: Yes

3. Have the authors made all data underlying the findings in their manuscript fully available?

Reviewer #1: Yes

Reviewer #2: Yes

4. Is the manuscript presented in an intelligible fashion and written in standard English?

Reviewer #1: Yes

Reviewer #2: Yes

5. Review Comments to the Author

Reviewer #1: Thank you for the opportunity to review the manuscript "Accuracy of COVID-19 relevant knowledge among youth: Do sources of information matter?". This article describes results from a survey of youth in grades 5-9 in a midwestern US school district to determine sources of information youth have about COVID and if the type or number of sources were associated with the accuracy of the knowledge.

Although the topic of this manuscript is of great importance, there are some major issues with the manuscript which need to be revised before I can recommend publication.

Introduction

First, much of the first paragraphs of the introduction section completely lacks citations - every sentence in an introduction section typically needs a citation and in its current form, it is very under cited.

Methods and Results

Additionally, I have major issues with the description of this as a mixed methods study - particularly the authors calling this a qualitative analysis - I understand that open-ended responses are often considered qualitative, but many qualitative researchers do not consider simply using a survey's open ended responses to be a form of qualitative research. Additionally, the author failed to describe their coding process - this needs much more description. The presentation of the "qualitative' results was also poorly presented - authors simply wrote a one sentence description and then listed some quotes under each section - this is not qualitative research and should not be called such.

Additionally, since this study does employ quantitative methods as well, I have major concerns about the 2% response rate. There is no description of how the survey questions were developed. Additionally, the sets of comics used should include additional information as this was a bit confusing to the reader.

There is very minimal information about how the survey was developed - I would have preferred additional information about the development of the questionnaire items.

Again, the presentation of the qualitative findings are not typical of qualitative presentation - just listing the items in quotes but not providing any context or description is not truly qualitative data analysis - it just appears that the researchers listed quotes from the open ended survey - this could be organized or summarized better. Additionally, there is no indication that a qualitative coding process took place - instead, authors just listed quotes under very short headings. This should, at a minimum, be listed as a major limitation of the study, but I would not classify what is included as true qualitative research.

Moreover, some of the presentation of the results are confusing and need to be re-organized. At times, I had trouble following the results section as so many findings are presented an they seem relatively haphazard. For example, why is SES presented in the middle of the results - this is typically presented at the start of the results section. I also had trouble following the presentation of the multivariate results and felt this could be organized/ presented in a clearer manner. The entire results section should also be written in past tense and much of it is written in present tense.

Discussion/Conclusion

I thought the discussion/ conclusion section was under conceptualized and more information comparing results to previous studies is needed.

Reviewer #2: COVID-19 has presented many questions regarding science communication and learning. The pandemic and large amount of misinformation about COVID-19 has demonstrated that we need better tools when teaching others about science and infections. Therefore, this study is extremely needed to better understand sources of information and their impact on science learning among youth. The authors do a great job of describing why this subject is important. However, at times the aims of this project and manuscript get lost in the literature review. Additional comments are provided below.

Abstract

• Opening the abstract with the projects guiding questions was unique and interesting. It grabbed my attention! However, the results presented, nor the conclusion statement seem to answer these questions. Specifically, the type and number of sources is not discussed. If these questions are not answered in the abstract, maybe they should be removed with more focus on what is known about the topic and gaps in scientific literature.

• It’s unclear why authors choose to include comics and how it related to the goal of the project.

• The abstract tends to focus more on the methods and provides little information regarding the results.

Introduction

• Lines 79 – 81: It would be helpful if the authors included additional information on how youth interpret information differently from adults.

• Lines 102 – 110: This paragraph presents more information about methods instead of background information relevant to the project.

• It would be nice if this section ended with a statement about how this manuscript or project helps to fill a gap in knowledge regarding youth and scientific information.

• One of the goals of this project was to examine the impact of comics on scientific learning. However, the authors provide limited information about why comics were chosen and what makes them advantageous compared to 10-minute news video mentioned in the results. Information on why comics were chosen, any background information about the impact of comics on science learning is needed.

Theoretical frameworks and literature review

• It is unclear why the authors are presenting an additional literature review. It would be helpful if this section focused more on the theoretical framework and how it guided this project.

• The way the authors currently have described the theoretical framework utilized for this project, it is unclear what constructs or principles were used for this project. More information about which elements of these theories was used to designing, analyzing, and/or reporting on this study is needed.

Study design and sample

• Lines 244 – 245: This sentence would be more appropriate in the analytic strategy since it mentions an analysis approach.

• Line 247: The authors describe 4 comic story conditions but in line 230, they describe the creation of three comic stories. If an additional comic story was created, this should be reflected in the earlier statement as well.

Concepts and measures

• This section is a little confusing because study and survey measures are combined with results.

• It would be helpful for the authors to write a results section that focuses on the results specific to the aims presented earlier in the paper.

Multivariate results

• This section discusses similar topics that were mentioned preciously in the concepts and measure section. To help with confusion and to streamline results, it might be helpful for the authors to present results related to each topic examined at once instead of repeating throughout. For example, the results from the multivariate models about number of information sources could be combined with the section focused on number of information sources that started at line 474.

Conclusion

• Additional citations are needed throughout the conclusion. For example lines 609-611, 618-620, and 625-626.

• It would be helpful if the discussion and conclusion incorporated constructs from the theories selected and how/if results from this study were supported by the theories selected.

• Line 621: the first “directly” can be deleted in this sentence discussing mainstream news media.

6. PLOS authors have the option to publish the peer review history of their article (what does this mean?). If published, this will include your full peer review and any attached files.

Reviewer #1: No

Reviewer #2: No

---

## [Author Response · Author response to Decision Letter 0]

30 Nov 2022

Manuscript ID PONE-D-22-11112 “Accuracy of COVID-19 relevant knowledge among youth: Do sources of information matter?”

Dear PLOS ONE Editors,

Thank you for this opportunity to revise our manuscript with the updated title, “Accuracy of COVID-19 relevant knowledge among youth: Number of information sources matters.” We found the comments from the editors and reviewers very helpful. The feedback has strengthened the paper, and we are grateful for the time and effort spent providing it. We organize our response here according to sections of the paper. We first list editor and reviewer feedback for each section, then provide an explanation of our revisions. 

We have included in our resubmission a clean revised paper and a version with tracked changes. Our responses to editors and reviewers are below. 

Abstract

Editors: “Per reviewer 2: Please address this review’s comments regarding the abstract.” Reviewer 2: “• Opening the abstract with the projects guiding questions was unique and interesting. It grabbed my attention! However, the results presented, nor the conclusion statement seem to answer these questions.

• It’s unclear why authors choose to include comics and how it related to the goal of the project. Specifically, the type and number of sources is not discussed. If these questions are not answered in the abstract, maybe they should be removed with more focus on what is known about the topic and gaps in scientific literature.

• The abstract tends to focus more on the methods and provides little information regarding the results.”

• We appreciate that the reviewer liked the way that we opened the abstract, which we have now revised to state and answer the questions we pose more clearly. We have revised the abstract to clearly indicate where we share results and where we answer our guiding questions that open the abstract. We discuss the type and number of sources. Of the 300-word abstract >150 words are dedicated to the results of study. We justify the use of comics in the introduction.

Introduction

Editors: “Per Reviewers 1 and 2: Please review the background and conclusions sections to ensure that the information provided is cited appropriately.”

“Per Reviewer 2: Please provide some context for why comics were chosen and any background, if applicable, from other studies that used comics to inform youth on science topics.”

Reviewer 1: “First, much of the first paragraphs of the introduction section completely lacks citations - every sentence in an introduction section typically needs a citation and in its current form, it is very under cited.”

Reviewer 2: “• Lines 79 – 81: It would be helpful if the authors included additional information on how youth interpret information differently from adults.

• Lines 102 – 110: This paragraph presents more information about methods instead of background information relevant to the project.

• It would be nice if this section ended with a statement about how this manuscript or project helps to fill a gap in knowledge regarding youth and scientific information.

• One of the goals of this project was to examine the impact of comics on scientific learning. However, the authors provide limited information about why comics were chosen and what makes them advantageous compared to 10-minute news video mentioned in the results. Information on why comics were chosen, any background information about the impact of comics on science learning is needed.”

• We have added citations to our introduction. We have now provided an introduction that includes a statement of the problem/gaps in literature on youth and adult health and science information acquisition and how it applies during the COVID-19 pandemic.

• We have included a paragraph on why comics are a unique form of media that have the potential to provide accurate, accessible, and equitable access for youth about science and health topics including COVID-19. 

• Per the journal guidelines and Reviewer 2, we have now added a statement at the end of the introduction that includes the aims of the study and general findings. 

Theoretical frameworks and background

Reviewer 2: “• It is unclear why the authors are presenting an additional literature review. It would be helpful if this section focused more on the theoretical framework and how it guided this project.

• The way the authors currently have described the theoretical framework utilized for this project, it is unclear what constructs or principles were used for this project. More information about which elements of these theories was used to designing, analyzing, and/or reporting on this study is needed.”

• We have rewritten the theory and background section so that it is not simply a second literature review following the introduction. The introduction is the statement of the problem and justification for comics. The theoretical frameworks and background section now highlights the important constructs from theory, prior research concerning those constructs, and how we used them to analyze and report on this study. 

Study design and sample

Editors: “Per Reviewer 1: Please provide additional information about the coding process for qualitative data in the methods section.”

Reviewer 1: “Additionally, I have major issues with the description of this as a mixed methods study - particularly the authors calling this a qualitative analysis - I understand that open-ended responses are often considered qualitative, but many qualitative researchers do not consider simply using a survey's open ended responses to be a form of qualitative research. Additionally, the author failed to describe their coding process - this needs much more description. The presentation of the "qualitative' results was also poorly presented - authors simply wrote a one sentence description and then listed some quotes under each section - this is not qualitative research and should not be called such.

Additionally, since this study does employ quantitative methods as well, I have major concerns about the 2% response rate. There is no description of how the survey questions were developed. Additionally, the sets of comics used should include additional information as this was a bit confusing to the reader.

There is very minimal information about how the survey was developed - I would have preferred additional information about the development of the questionnaire items.”

Reviewer 2: “This section is a little confusing because study and survey measures are combined with results.”

• We are including a second deidentified dataset in Excel that includes all open-ended questions and codes in Zenodo in addition to the quantitative dataset. 

• We have clarified the coding process.

• We now make clear that this is not a mixed-methods study. We coded open-ended survey responses into thematic categories to facilitate quantitative analyses. Because most of the responses were straightforward (e.g., "news”), there were minimal coding decisions. The first author (Hill) and an undergraduate did the first round of categorization of the original codes into fewer thematic categories. When there were questions, they brought those questions to the team to condense categories. For example, some people said "CDC" or "Fauci," and we collapsed these to "experts". 

• There was one longer comic (10 pages) and two smaller comics (five pages) in this series of comics. To create equal conditions in the randomly assigned survey, Comic I was separated into two five-page comics. Therefore, we had four randomized comic conditions that were five pages long each. Although Comic I was 10 pages, the narrative did not appear as disjointed to youth due to formatting and headings that differentiated the first five pages from the second five pages. Additionally, the information presented to youth was different between the two five-page sections in Comic I, so we were able to formulate different knowledge-based questions across these two conditions. We have added a clearer explanation of this in the paper.

• The 2% response rate is unfortunate. We did send out the survey invitation several times. We were also struggling with contextual challenges: this study was conducted during the pandemic, and we were required by the school district to first obtain consent from parents, then have parents connect students to the survey by passing their device to their student, sending their student a unique code, or providing a contact email for consented students. The school district would not let us contact students directly prior to parental consent. We needed the parents to complete a brief consent form, then invite their child to do the survey. We believe under different conditions that the response rate would have been higher since previous in-school studies in the same district garnered response rates over 60%. We are careful to not over-generalize from the results. We do point out that this study involves an experiment, and therefore the randomization in part addresses the limits of the low response rates. We have added a description of how survey items were developed, and we have cited the comics and provided a URL link to the study that describes them in more detail for the reader although the link must be redacted for blind review.

• We have reorganized the entire paper, so that all results are in the results section and not interspersed in the measures. Because this was a complex study with multiple stages for variables creation, it initially made conceptual sense to take the reader through that process. We understand that this may have been confusing, so we appreciate the opportunity to reorganize the paper. 

Results

Editors: “Per Reviewers 1 and 2: Please modify the results section to link the results to the aims of the study and to clarify the findings.”

Reviewer 1: “Again, the presentation of the qualitative findings are not typical of qualitative presentation – just listing the items in quotes but not providing any context or description is not truly qualitative data analysis – it just appears that the researchers listed quotes from the open ended survey – this could be organized or summarized better. Additionally, there is no indication that a qualitative coding process took place – instead, authors just listed quotes under very short headings. This should, at a minimum, be listed as a major limitation of the study, but I would not classify what is included as true qualitative research.

Moreover, some of the presentation of the results are confusing and need to be re-organized. At times, I had trouble following the results section as so many findings are presented an they seem relatively haphazard. For example, why is SES presented in the middle of the results – this is typically presented at the start of the results section. I also had trouble following the presentation of the multivariate results and felt this could be organized/ presented in a clearer manner. The entire results section should also be written in past tense and much of it is written in present tense.”

Reviewer 2: “It would be helpful for the authors to write a results section that focuses on the results specific to the aims presented earlier in the paper. This section discusses similar topics that were mentioned preciously in the concepts and measure section. To help with confusion and to streamline results, it might be helpful for the authors to present results related to each topic examined at once instead of repeating throughout. For example, the results from the multivariate models about number of information sources could be combined with the section focused on number of information sources that started at line 474.”

• As described in a previous response, this is not a mixed methods paper and we have clarified our process for coding open-ended questions from the survey. 

• We have reorganized the entire paper so that all results are in the results section.

• We have reorganized the results and organized them by research question. 

• Per Reviewer 1, we have revised the results section to be in the past tense throughout.

Discussion and conclusion

Editors: “Per Reviewers 1 and 2: Please review the background and conclusions sections to ensure that the information provided is cited appropriately.”

Reviewer 1: “I thought the discussion/ conclusion section was under conceptualized and more information comparing results to previous studies is needed.”

Reviewer 2: “Additional citations are needed throughout the conclusion. For example lines 609-611, 618-620, and 625-626. It would be helpful if the discussion and conclusion incorporated constructs from the theories selected and how/if results from this study were supported by the theories selected.”

• We have rewritten the conclusion to tie back into the introduction and to cite prior literature as it pertains to our results. We have added multiple additional citations. 

Again, we thank the reviewers and editors for this opportunity to revise this manuscript. We are happy to make any additional changes or to answer further questions.

---

## [Editor Report · Decision Letter 1]

6 Dec 2022

Accuracy of COVID-19 relevant knowledge among youth: Number of information sources matters

PONE-D-22-11112R1

Dear Dr. Hill,

We’re pleased to inform you that your manuscript has been judged scientifically suitable for publication and will be formally accepted for publication once it meets all outstanding technical requirements.

Kind regards,

Cindy Prins

Academic Editor

PLOS ONE
---

## [Editor Report · Acceptance letter]

13 Dec 2022

PONE-D-22-11112R1 

Accuracy of COVID-19 relevant knowledge among youth: Number of information sources matters 

Dear Dr. Hill:

I'm pleased to inform you that your manuscript has been deemed suitable for publication in PLOS ONE. Congratulations! Your manuscript is now with our production department. 

Kind regards, 

on behalf of

Dr. Cindy Prins 

Academic Editor

PLOS ONE